# xRAG: Extreme Context Compression for Retrieval-augmented Generation with One Token

Xin Cheng[1]*   Xun Wang[2]   Xingxing Zhang[2]   Tao Ge[2]
Si-Qing Chen[2]   Furu Wei[2]   Huishuai Zhang[1,3]   Dongyan Zhao[1,3]

[1] Peking University    [2] Microsoft
[3] National Key Laboratory of General Artificial Intelligence
chengxin1998@stu.pku.edu.cn

## Abstract

This paper introduces xRAG, a novel context compression method designed specifically for retrieval-augmented generation. xRAG redefines the use of document embeddings in dense retrieval—traditionally limited to retrieval purposes—by integrating them as features from the retrieval modality. Through a modality fusion approach, xRAG effectively merges these embeddings into the language model's representation space, eliminating the need for their textual counterparts and achieving an extreme compression rate. In xRAG, the modality bridge is the only trainable component, while the retriever and language model remain frozen. This design choice allows for the reuse of offline-constructed document embeddings and preserves the plug-and-play nature of retrieval augmentation. Experimental results demonstrate that xRAG achieves an average improvement of over 10% across six knowledge-intensive tasks, compatible with various language model backbones, ranging from a dense 7B model to an 8x7B Mixture of Experts configuration. xRAG not only significantly outperforms previous context compression methods but also matches the performance of uncompressed models on several benchmarks, while reducing overall FLOPs by a factor of 3.53. This work pioneers new avenues in retrieval-augmented generation through multimodal fusion, potentially setting a groundwork for future developments in efficient and scalable retrieval systems.

## 1 Introduction

Retrieval-Augmented Language Models (RALMs) [42, 22, 9, 14, 70] have shown exceptional performance in a variety of knowledge-intensive tasks. By retrieving domain-specific, long-tailed, and up-to-date knowledge from a non-parametric datastore, RALMs significantly extend the boundaries of parametric Large Language Models (LLMs). However, the integration of entire documents into prompts can significantly increase inference costs and may surpass the context limit of LLMs [29, 79]. As illustrated in Figure 1, while the inclusion of a relevant document enables the LLM to generate accurate responses, it does so at the expense of processing documents that expand the original query by more than tenfold.

How might we mitigate the costs associated with extended context while maintaining the benefits of retrieval augmentation? Recent research interest has converged on a promising direction: Context Compression. This concept is pursued through two primary strategies: soft-prompting methods, such as Gist [59], AutoCompressor [15], and ICAE [20], which compress the context into dense memory slots, and hard-prompting methods, such as LLMLingua [29] and RECOMP [79], where

---

*Work done during internship at Microsoft, corresponding to Xun Wang, Huishuai Zhang and Dongyan Zhao

38th Conference on Neural Information Processing Systems (NeurIPS 2024).

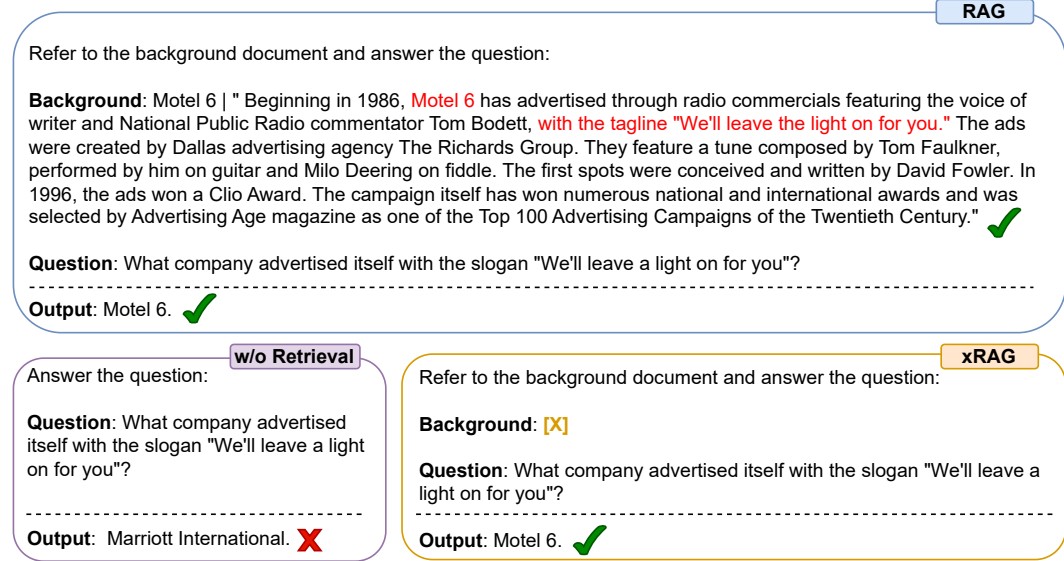

Figure 1: xRAG enables efficient retrieval augmentation by adding one document token [X].

compression is applied on the surface form. These approaches, however, either require significant memory for storing LLM activations (e.g., 1.05 MB per token as reported by [59]) or suffer from relatively low compression rates. More critically, these methods overlook a crucial characteristic of RALMs: through large-scale contrastive learning with question-document pairs, modern dense retrieval systems already distill document content into a single high-dimensional embedding, and *this embedding reveals (almost) as much information as the text* [58, 44].

In this paper, we pioneer an innovative approach to retrieval augmentation and context compression through the lens of modality fusion. Drawing from multimodal research, where text-only language models are taught to "perceive" and "listen," a pretrained modality encoder like CLIP [64] is typically used to extract modality features. These features are then integrated into language models using a modality fusion bridge [45, 52, 34]. Building on the conceptual overlap between the retriever encoder and modality encoder, we introduce xRAG. This model redefines document embeddings from dense retrieval—traditionally solely for retrieval purposes—as retrieval modality features. xRAG employs a modality fusion methodology to seamlessly integrate these embeddings into the language model's representation space, thus obviating the need for textual counterparts and achieving significant context compression. In xRAG, the modality bridge is the only trainable component, while both the retriever and the LLM are kept frozen. This design decision facilitates the reuse of pre-constructed document embeddings and maintains the plug-and-play nature of retrieval augmentation—two essential factors for a functional RAG system.

To verify the effectiveness and versatility of our framework, we conducted comprehensive experiments with different LLM backbones, ranging from a dense 7B model to an 8x7B Mixture of Experts model. Our results reveal that adding just one document token could lead to over a 10% improvement across six knowledge-intensive tasks, significantly surpassing previous compression methods. xRAG also delivers results comparable to uncompressed models on several benchmarks. This is remarkable considering that the only trainable component constitutes less than 0.1% of the LLM's parameters. In terms of efficiency, xRAG reduces total FLOPs by a factor of 3.53 compared to the uncompressed RAG model. We further provide detailed analyses of xRAG, examining various training strategies, data blends, and component selections for the retrieval system. We believe this research sets a strong foundation for the development of future efficient and scalable retrieval-augmented systems.

## 2    Related Work

**Retrieval-augmented Generation**    Equipping a parametric language model with a non-parametric datastore has proven effective for a range of NLP tasks, including language modeling [37, 57, 87],

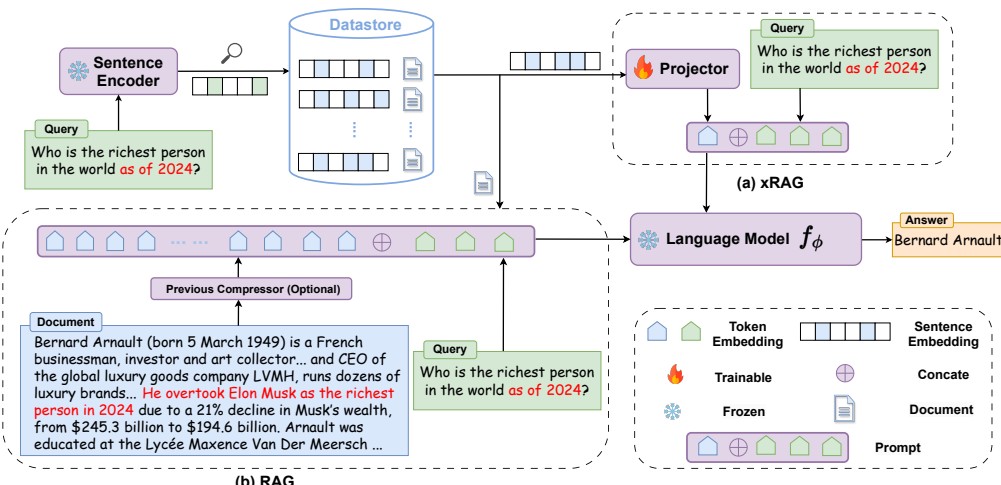

Figure 2: Overview of xRAG (a) and RAG (b). For a given query, RAG typically concatenates the retrieved document with the query, significantly extending the context length. In contrast, xRAG addresses this issue through modality fusion by directly projecting the document embedding into the LLM's representation space. This allows for efficient retrieval-augmentation with the addition of only one token.

open-domain question answering [25, 42, 70, 86], domain adaptation [7] and machine translation [36, 12], among others. Given the vast design space of this generation paradigm, numerous approaches with different focuses have been proposed. For instance, RETRO [9] and PlugLM [13] introduce architectural innovations for enhanced integration with the non-parametric datastore. REALM [22] pioneers an end-to-end approach for simultaneous optimization of the language model and retriever. REPLUG [70] and RA-DIT [51] improve retriever alignment using feedback from LLMs. DSP [38] and InteR [18] investigate complex interactions between the retriever and the language model. Selfmem [14] utilizes a reward model to refine retrieval and generation iteratively. Self-RAG [4] incorporates a self-reflection mechanism to enhance the quality and factuality of language model outputs. For a detailed overview, see [19, 5, 3]. Our contribution, xRAG, stands out by implementing a modality fusion approach to retrieval augmentation, creating an effective and efficient RAG system.

**Context Compression**    Context compression, aimed at reducing the input length for LLMs while retaining essential information, has recently attracted substantial interest [47]. Gist [59] achieves a compression rate of up to 26x by modifying the attention mask and caching soft gist token activations. ICAE [20]AutoCompressor [15], and 500xCompressor[48] condense lengthy contexts into succinct, compact memory slots, which are directly utilizable by LLMs for diverse functions. LLMLingua [29, 30, 62] and CompAct[82] introduces a coarse-to-fine prompt compression technique based on perplexity scores and distilled token-level score. While these methods are generally applicable, others are tailored specifically for RAG systems, such as FilCo [76] and RECOMP [79]. A concurrent work directly employs passage embeddings for efficient listwise reranking [53]. For an in-depth comparison of these compression methods regarding memory efficiency, compression rates, and adaptability, refer to Appendix A.

## 3 Methods

**Problem Formulation**    In retrieval-augmented generation, a non-parametric datastore $\mathbb{D} = \{(\mathrm{E}_i, \mathrm{D}_i)\}_{i=1}^{|\mathbb{D}|}$ consists of pairs where each $\mathrm{D}_i$ represents a document chunk as a sequence of $L_i$ tokens $\mathrm{D}_i = \{d_1^i, \ldots, d_{L_i}^i\}$. Correspondingly, $\mathrm{E}_i$ is the dense representation derived from a sentence embedding model $\mathbf{SE}_\theta(\cdot)$ with input $D_i$. For an input query $q$, its dense representation $\mathbf{SE}_\theta(q)$ is used to find the relevant documents by matching against the collection $\{\mathrm{E}_i\}_{i=1}^{|\mathbb{D}|}$ with certain similarity search algorithm such as MIPS. After retrieval, the system selects a relevant pair $(\mathrm{E}, \mathrm{D})$ from $\mathbb{D}$, concatenates the chosen document D with $q$, and processes the combined input with a language model

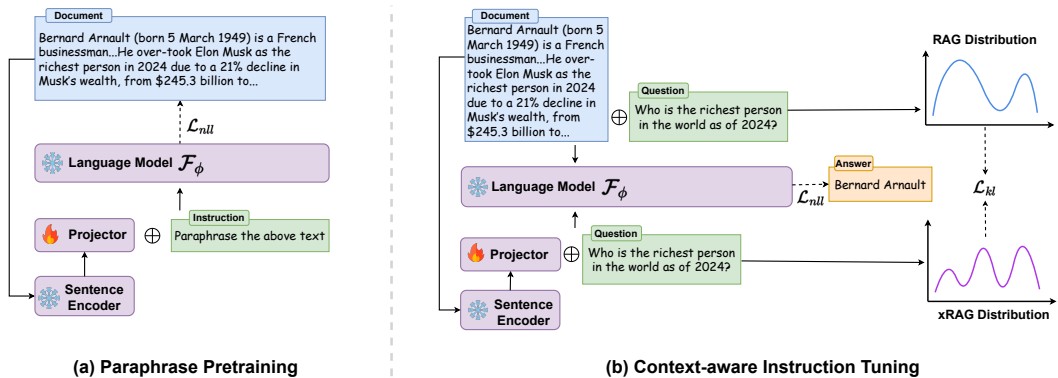

Figure 3: Two-stage training strategy of xRAG including (a) Paraphrase Pre-training on unlabeled corpus and (b) Context-aware Instruction Tuning optimized with labeled data and self-distillation.

$\mathcal{F}_\phi(\mathrm{D} \oplus q)$. Optionally, a context compression module $\mathcal{C}$ can be integrated to reduce the length of D from $L$ to a more concise $l$, achieving a compression ratio of $\frac{L}{l}$.

## 3.1 xRAG Architecture

Traditional methods for document compression typically focus on surface form of the document [29, 76, 79]. In contrast xRAG tackle the problem from a modality fusion view. Concretely, we introduce a modality projector $\mathbf{W}$, which is trained to directly project the retrieval features E into the LLM representation space. Our proposed framework is visually contrasted with the traditional RAG system in Figure 2. In the standard RAG, the input to the LLM comprises the embeddings $\mathrm{Emb}(\mathrm{D} \oplus q)$ of length $|\mathrm{D}| + |q|$, where $\mathrm{Emb}$ signifies the embedding layer of the LLM. Conversely, with xRAG, the modified input is represented as $\mathbf{W}(\mathrm{E}) \oplus \mathrm{Emb}(q)$, which yields a substantially reduced length of $1 + |q|$. In this framework, the challenges come from the modality fusion: How can a text-only language model understand features from retrieval modality? To achieve this, we explore a two-stage training strategy: Paraphrase Pretraining followed by Context-aware Instruction Tuning.

## 3.2 Paraphrase Pretraining

Similar to the pretraining strategies employed in vision-language models that use image-captioning data to align two modalities [52, 16, 54], the primary objective of our paraphrase pretraining is to build a compatible representation between the extracted retrieval feature and the corresponding document. Illustrated in Figure 3(a), for each pair $(\mathrm{E}, \mathrm{D})$ in a retrieval corpus $\mathbb{D}$, we employ a natural language instruction $\mathbf{X}_{\texttt{instruct}}$ to prompt the LLM to undertake a paraphrasing task (e.g. "[X] The above text could be paraphrased as: [D]", where [X] and [D] are placeholders for $\mathbf{W}(\mathrm{E})$ and document D)[2]. In this setup, the model learns to connect $\mathbf{W}(\mathrm{E})$ and D by recovering D on the condition of $\mathbf{W}(\mathrm{E})$ and the model is optimized by:

$$\mathcal{L}_{\mathrm{nll}} = -\sum_{i=1} \log p_\phi(d_i | \mathbf{W}(\mathrm{E}), \mathbf{X}_{\texttt{instruct}}, d_{<i}) \tag{1}$$

where $p_\phi$ is given by the softmax distribution of LLM $\mathcal{F}_\phi$, and $d_{<i}$ denotes the document token before current prediction token $d_i$, achieved by casual attention mask in auto-regressive LMs.

## 3.3 Context-aware Instruction Tuning

After the pretraining phase, although the language model $\mathcal{F}_\phi$ has developed an internally compatible representation, it has never been explicitly trained to utilize these features for downstream tasks. To address this gap, we proceed to instruct the model in harnessing the fused feature $\mathbf{W}(\mathrm{E})$ by continually training the model on data where the answer is closely associated with the given context, including reading comprehension, summarization, and open domain question answering data. We

---

[2]To maintain diversity, we sample from an instruction pool, which could be found in Appendix B.

constructed an mixed dataset, containing approximately 1 million entries from open-source datasets, as detailed in Appendix C. For each triplet in the dataset, $(\mathbf{X}_{\texttt{context}}, \mathbf{X}_{\texttt{question}}, \mathbf{X}_{\texttt{answer}})$, we initially obtain the sentence representation for $\mathbf{X}_{\texttt{context}}$ via the embedding model $\mathrm{E}_{\texttt{context}} = \mathbf{SE}_\theta(\mathbf{X}_{\texttt{context}})$. Subsequently, we refine the optimization of on two directions:

**Optimization I: Language Modeling.** Aligned with established instruction tuning methodologies [75, 24, 51, 43], our objective is to finetune the model so that it generates the correct output when provided with a specific instruction, conditioned upon the given context information. Unlike traditional models that utilize the textual context $\mathbf{X}_{\texttt{context}}$, our method employs a dense feature $\mathrm{E}_{\texttt{context}}$ to encapsulate the context information:

$$\mathcal{L}_{\texttt{nll}} = -\sum_{i=1} \log p_\phi(\mathbf{X}_{\texttt{answer},i} | \mathbf{W}(\mathrm{E}_{\texttt{context}}), \mathbf{X}_{\texttt{question}}, \mathbf{X}_{\texttt{answer},<i}) \tag{2}$$

**Optimization II: Self-Distillation.** The second trajectory of optimization aims to guide the xRAG in the effective utilization of contextual information, drawing from the principles of self-distillation [2, 71] and imitation learning [61, 23]. By considering the RAG model as a "teacher" and xRAG as a "student", we endeavor to distill the knowledge from RAG, thereby enabling xRAG to emulate the RAG model's proficiency in handling the full, uncompressed documents. This approach enhances xRAG's resilience in scenarios where it encounters noisy or irrelevant context that may not directly lead to the correct answer, detailedly discussed in § 6.1. Concretely, for a language model $\mathcal{F}_\phi$ using either $\mathbf{X}_{\texttt{context}}$ or $\mathrm{E}_{\texttt{context}}$ as the source of context, our objective is to minimize the divergence between the two resulting output distributions. This discrepancy is measured using the Kullback-Leibler (KL) divergence:

$$\mathcal{L}_{\texttt{kl}} = D_{\mathrm{KL}}(p_\phi(\mathbf{X}_{\texttt{answer}} | \mathbf{X}_{\texttt{context}}, \cdot) \, || \, p_\phi(\mathbf{X}_{\texttt{answer}} | \mathbf{W}(\mathrm{E}_{\texttt{context}}), \cdot)) \tag{3}$$

Here $\mathbf{X}_{\texttt{question}}$ is omitted for brevity and the final loss is the linear combination controlled by a hyperparameter: $\mathcal{L}_{nll} + \alpha \mathcal{L}_{kl}$.

### 3.4 Design Principle

In designing the projector $\mathbf{W}$, our primary objective is to maintain the simplicity of the framework. We therefore opted for a two-layer MLP while other more sophisticated module such as Q-Former [45] could also be considered. Notice that the projector is the only trainable component, accounting for only 0.46% of the total parameters in the Mistral-7b model and 0.07% in the Mixtral-8x7b model. Such a design choice departs from previous studies that necessitated full-parameter tuning to adapt LLMs for compressed contexts [59, 76, 15]. We believe this approach will likely be more accessible and practical because, fundamentally, the RAG itself functions as a plug-and-play module for LLMs, and so should its compressed version. This design also avoid the risk of compromising other core capabilities of LLM during full-parameter tuning, as observed in [55, 56].

Moreover, in contrast to other compression methods that necessitate storing LLM activations for each compressed token [59, 20, 15]—an impractical strategy in the RAG setting, given the millions of documents involved—our method introduces no additional memory overhead. Instead, it leverages offline-constructed document embeddings, originally designed for retrieval. To summarize, xRAG not only simplifies the integration process but also avoids unnecessary computational or memory expenses.

## 4 Experimental Setup

### 4.1 Evaluation Dataset

We evaluated the performance of xRAG primarily on knowledge-intensive tasks which encompass a range of challenges: (1) three Open Domain Question Answering datasets that address questions on a wide array of topics: Natural Questions [41], TriviaQA [33], and Web Questions [8]. (2) one Multihop Question Answering dataset, HotpotQA [81], which necessitates multi-step reasoning to generate answers. (3) one Long-form Question Answering dataset, TruthfulQA [50], that requires the generation of long-form and truthful responses. (4) one fact-checking dataset, FactKG [39], which challenges the model to use complex reasoning to determine the factual accuracy of given claims.

In line with the KILT [63] and GenRead [84], we assessed three ODQA datasets and HotpotQA using the Exact Match (EM) metric, FactKG with Accuracy, and for the long-form QA, we used both the F1 score and Rouge-L (R-L) score. These tasks demand a broad spectrum of world knowledge and have been extensively explored in the retrieval-augmentation literature [42, 22, 35, 10, 25, 70].

### 4.2 Implementation Details

To demonstrate the versatility of our framework, we choose two backbones, differing in scale and architecture: Mistral-7b [27] and Mixtral-8x7b [28]. For the retrieval corpus, we utilized the Wikipedia dump from December 2021, which was pre-processed into passages following the methodology described in [26]. This resulted in approximately 37 million passages, each averaging 180 tokens in length. Our default retrieval model is the SFR [1], which, at the time of writing this paper, holds the leading position on the MTEB leaderboard [60]. We use *top-1* ranked document for inclusion in our instruction-tuning dataset and for the evaluation of downstream tasks. More details are provided in Appendix D. Code is available at: `https://github.com/Hannibal046/xRAG`.

### 4.3 Baselines

In determining appropriate baselines for comparison, we adhered to a fundamental principle: the selected compression methods must support the general plug-and-play capability of retrieval augmentation. This entails that they should function effectively without the need for dataset-specific tuning [79, 76], or any alteration to the parameters of LLMs [76, 59]. Furthermore, given the extensive volume of the retrieval corpus, it is essential that these compression methods demonstrate memory efficiency, specifically by not requiring the storage of LLM activations for each individual token [59, 20, 15]. With these criteria in mind, our evaluation compares xRAG to the following baselines: **(I)** Primarily, we consider two variants of LLMs: one that operates without retrieval augmentation and another that includes it. These serve as the lower and upper performance bounds for our study of compression techniques, respectively. **(II)** Additionally, our comparisons extend to LLMLingua [29], a plug-and-play approach for context compression. **(III)** Taking inspiration from [59], we incorporate a method of discrete compression using TF-IDF. This approach yields compression rates comparable to those achieved by xRAG and serves as the lower bound for discrete compression.

Table 1: Experimental results on six downstream tasks. The best results are in **bold** and the second best are with underscore. Percentage in the brackets denotes the relative improvement over non-retrieval setting. LLMs are frozen during the experiments and retrieved documents are set the same for different compression methods. ‡ and † denotes different compression ratio.

| Task Type | NQ | TriviaQA | WebQA | HotpotQA | TrutufulQA | | FactKG | Average | # Doc Length |
|---|---|---|---|---|---|---|---|---|---|
| | | Open-Domain QA (EM) | | Multihop QA (EM) | Long-form QA (F1/R-L) | | Fact Checking (Acc) | | |
| **Mistral-7b** | | | | | | | | | |
| w/o retrieval | 30.25 | 57.08 | 34.89 | 27.02 | 26.23 | 25.51 | 54.78 | 36.54 (0.0%) | 0 |
| w retrieval | **42.71** | **65.88** | 37.84 | **38.79** | 26.50 | 25.92 | **67.76** | **43.63** (19.4%) | 175.1 |
| ***with Compression** | | | | | | | | | |
| LLMLingua † | 30.64 | 57.94 | 32.63 | 29.91 | 25.70 | 25.10 | 64.17 | 38.01 (4.0%) | 98.6 |
| LLMLingua ‡ | 28.81 | 57.09 | 32.33 | 29.13 | 26.10 | 25.39 | 63.57 | 37.48 (2.5%) | 61.1 |
| TF-IDF | 30.25 | 58.49 | 35.43 | 26.62 | 26.33 | 25.83 | 59.56 | 37.49 (2.6%) | 1 |
| xRAG | 39.10 | 65.77 | **39.40** | 34.05 | **28.10** | **27.71** | 63.08 | 42.46 (16.2%) | 1 |
| **Mixtral-8x7b** | | | | | | | | | |
| w/o retrieval | 41.99 | 71.10 | 40.31 | 32.87 | 25.60 | 24.90 | 62.64 | 42.76 (0.0%) | 0 |
| w retrieval | 45.15 | 70.34 | 41.26 | **43.46** | 27.10 | 25.80 | **70.42** | 46.22 (8.0%) | 175.1 |
| ***with Compression** | | | | | | | | | |
| LLMLingua† | 37.65 | 67.70 | 36.02 | 35.66 | 25.99 | 25.39 | 67.98 | 42.32 (-1.0%) | 96.6 |
| LLMLingua‡ | 37.81 | 67.81 | 35.78 | 35.27 | 25.68 | 25.00 | 68.03 | 44.17 (-1.3%) | 61.1 |
| TF-IDF | 41.19 | 69.94 | 41.63 | 32.05 | 26.80 | 26.00 | 66.17 | 43.41 (1.4%) | 1 |
| xRAG | **47.28** | **74.14** | **44.50** | 39.66 | **27.80** | **26.64** | 68.20 | **46.91** (9.7%) | 1 |

## 5 Experimental Results

### 5.1 Knowledge Intensive Tasks

In Table 1, we present our main results. Across both Mistral-7b and Mixtral-8x7b configurations, we observe a consistent and significant improvement when retrieval augmentation is applied (p-value

< 0.05), although the gains are more modest for the larger model configurations. This trend aligns with observations reported by [70, 51]. Further analysis on the efficacy of various compression techniques reveals that xRAG outperforms other approaches by a large margin. Remarkably, xRAG not only reduces the token count drastically—from 175.1 to a single token—but also maintains robust performance levels. In some instances, xRAG's performance is comparable to, or even exceeds, that of the uncompressed models. Specifically, in the Mistral-7b configuration, xRAG achieves nearly the same performance improvement as the uncompressed model (16.6% compared to 19.4%), and in the Mixtral-8x7b configuration, it surpasses the uncompressed model (9.7% compared to 8.0%). One possible reason lies in the vulnerability of current RAG system when the irrelevant or misleading documents are presented, a topic detailed discussed in §6.1. We also observe that xRAG performs well in tasks that require document understanding, such as TriviaQA. However, in tasks that demand reasoning over document, like HotpotQA and FactKG, xRAG lags behind by a considerable gap.

Table 2: Comparison of RAG and xRAG performance in CUDA Time and GFLOPS.

| | CUDA Time (ms) | | | GFLOPs | | |
| | RAG | xRAG | Improvement | RAG | xRAG | Improvement |
| --- | --- | --- | --- | --- | --- | --- |
| FactKG | 431.5 | 215.6 | x2.01 | 4683.8 | 1289.5 | x3.63 |
| NQ | 918.7 | 611.3 | x1.51 | 1338.6 | 384.0 | x3.48 |
| TriviaQA | 807.1 | 512.1 | x1.57 | 1667.2 | 492.3 | x3.38 |
| WebQA | 872.6 | 577.3 | x1.51 | 1405.1 | 386.8 | x3.63 |
| Average | | | **x1.64** | | | **x3.53** |

## 5.2 Computational Efficiency

In this section, we conduct a thorough assessment of our framework's computational efficiency and memory management. To rigorously evaluate our model, we employed `Torch Profiler`[3] to measure the CUDA Time (milliseconds) and Giga FLOPs of both the RAG and xRAG models across four real-world datasets. In these evaluations, the Mistral-7b, operating in bfloat16 inference mode, served as the base LLM. CUDA Time and GFLOPs were calculated on an average per batch basis with a fixed batch size, and GFLOPs were normalized by the number of generated tokens. These experiments were performed on the same computational hardware, specifically an Nvidia A100 and an AMD EPYC 7V12 64-Core Processor. As depicted in Table 2, despite variations in prompt and generation lengths across the datasets, xRAG significantly outpaced the RAG model, achieving a x1.64 increase in CUDA Time efficiency and a x3.53 reduction in GFLOPs.

# 6 Analysis

## 6.1 Evaluation Beyond the Overall Score

Although retrieval augmentation generally boosts performance as shown by aggregate metrics, it may not uniformly benefit all instances. In certain cases, the retrieval system might provide irrelevant or even misleading information, leading to incorrect answers that were previously correct [83, 76, 5]. To enable a more fine-grained evaluation, we introduce two novel metrics: the Resilience Rate and the Boost Rate. The resilience rate quantifies the percentage of instances in which the system's responses remain correct both before and after retrieval augmentation, highlighting the system's stability and robustness. Conversely, the boost rate measures the percentage of instances that were initially answered incorrectly but were rectified following the introduction of a retrieved document, thereby assessing the efficacy of retrieval augmentation. An ideal RAG system should have both high resilience rate and boost rate.

In Figure 4, we display these metrics for the uncompressed RAG and two compression methods: LLMLingua and xRAG. Surprisingly, although retrieval augmentation generally enhances performance, the resilience rate for RAG averages only 75.2%, indicating that retrieval can adversely affect about one-quarter of previously correct answers. In contrast, xRAG demonstrates considerable

---

[3]`https://pytorch.org/docs/stable/profiler.html#module-torch.profiler`

robustness across all evaluated datasets. This robustness largely stems from xRAG's ability to maintain an unbiased stance toward the internal knowledge representation of the LLM, especially when confronted with noisy retrieval content. Similar trends are noted in [51, 55], where search-augmented instruction learning is shown to bolster the robustness of language models. However, xRAG still lags behind RAG in boost rate, particularly in multi-hop reasoning tasks. It is crucial to note that a high resilience rate does not necessarily mean that the LLM disregards the provided information, which could potentially lead to a reduced boost rate. A comparative analysis with LLMLingua indicates that xRAG is not only more robust but also more effective.

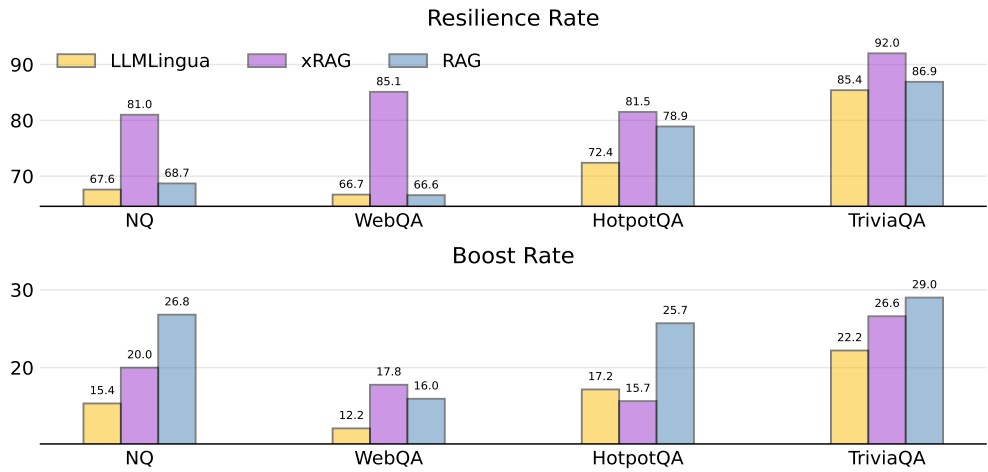

Figure 4: Resilience rate and boost rate of three augmentation methods: LLMLinuga, xRAG and RAG over a Mixtral-8x7b baseline without retrieval augmentation.

## 6.2 What makes xRAG effective?

Table 3: Ablation on different training strategy for xRAG.

| | NQ | TriviaQA | WebQA | HotpotQA | Averaged Performance | Resilience | Boost |
|---|---|---|---|---|---|---|---|
| **Mistral-7b** | | | | | | | |
| xRAG | 39.10 | 65.77 | 39.40 | 34.05 | **44.58** | 82.3% | **22.2%** |
| w/o finetune | 30.14 | 59.48 | 35.19 | 26.70 | 37.87 | 66.6% | 20.8% |
| w/o pretrain | 31.25 | 59.07 | 41.19 | 24.32 | 38.95 | 79.8% | 14.1% |
| w/o nll | 35.46 | 65.27 | 39.57 | 31.80 | 43.02 | **83.7%** | 19.4% |
| w/o self-kd | 34.99 | 64.33 | 39.22 | 27.45 | 41.49 | 76.2% | 20.8% |
| w LoRA | 35.71 | 60.14 | 40.45 | 22.91 | 39.80 | 76.0% | 18.0% |
| **Mixtral-8x7b** | | | | | | | |
| xRAG | 47.48 | 74.14 | 44.50 | 39.66 | **51.45** | 84.9% | 20.0% |
| w/o finetune | 34.46 | 64.08 | 34.89 | 30.43 | 40.96 | 65.9% | 17.8% |
| w/o pretrain | 42.54 | 71.17 | 47.44 | 31.23 | 48.09 | 85.0% | 14.2% |
| w/o nll | 45.10 | 72.85 | 45.03 | 37.11 | 50.02 | 84.8% | 18.9% |
| w/o self-kd | 42.38 | 72.26 | 44.73 | 32.41 | 47.94 | 79.8% | 18.9% |

This section delves into a thorough evaluation of various elements that contribute to xRAG's overall performance, focusing on its training strategy, the blend of datasets used and the effect of different embedding models. Due to the space limit, we present the last factor in Appendix E.

**1. Training Strategy** We carefully ablate four optimization choices: pretraining, instruction tuning, and two optimization objectives—language modeling (nll) and self-distillation (self-kd). We also train a Mistral-7b with LoRA on our instruction tuning dataset to rule out the possibility that our improvement simply comes from tuning on more data. The outcomes are presented in Table 3. Our analysis reveals that the interplay of different training strategies significantly contributes to the

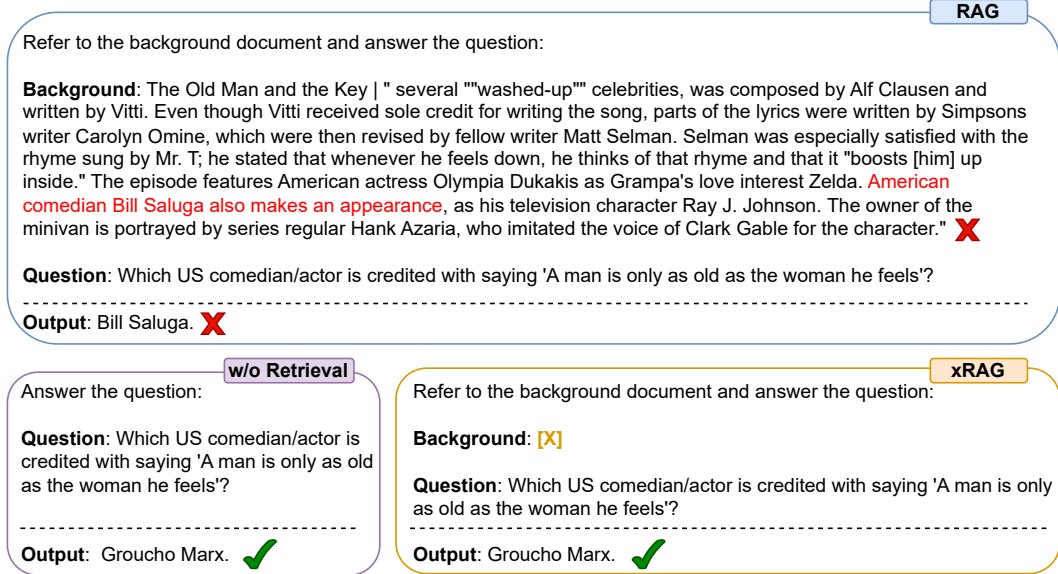

Figure 5: Given the misleading document, RAG model tend to generate a wrong answer based on the document, while xRAG demonstrate its robustness by leveraging the internal knowledge of LLM.

performance of our framework. In the case of Mistral-7b, pretraining and finetuning phases are of equal significance to the end results. However, for Mixtral-8x7b, the impact of pretraining is notably diminished, likely due to the larger model's enhanced capability to incorporate multi-modality information. Furthermore, we find that during finetuning, self-distillation is more important than language modeling. The primary advantage of self-distillation lies in bolstering the resilience rate of the xRAG system. Optimization with nll loss tends to cause an overreliance on context information, rendering the system more vulnerable when the retriever fails to fetch relevant documents.

Table 4: Abaltion results on different data selection strategy.

| | # Train | NQ | TriviaQA | WebQA | HotpotQA | Average Performance | Resilience | Boost |
|---|---|---|---|---|---|---|---|---|
| **xRAG (Mistral-7b)** | 955k | 39.10 | 65.77 | 39.40 | 34.05 | **44.58** | 82.3% | 22.2% |
| w RC only | 488k | 36.98 | 65.77 | 41.39 | 32.82 | 44.24 | 81.9% | 22.4% |
| w QA only | 385k | 36.45 | 65.57 | 41.14 | 31.80 | 43.74 | 80.5% | 22.1% |
| w Summ only | 81k | 36.37 | 64.95 | 40.40 | 31.98 | 43.42 | 78.8% | 22.8% |

**II. Instruction-tuning Dataset Blend** As discussed in §3.3, our instruction-tuning dataset primarily comprises three categories: reading comprehension, open-domain QA, and text summarization. To explore the effects of different data blends, we instruction-tune three xRAG model variants, each using data from these distinct categories. The results are shown in Table4. Our analysis reveals that among the dataset blends, reading comprehension data most significantly enhances the xRAG model's performance, as evidenced by both high resilience and boost rates. Intriguingly, when tuned solely with summarization data, xRAG still manages to deliver strong performance on QA datasets it has never been exposed to. This finding underscores that the advantages of instruction tuning for xRAG are not rooted in task-specific knowledge. Instead, they derive from the model's improved ability to utilize projected context information effectively.

### 6.3 Case Study

In Figure 5, we show one interesting case about the robustness of xRAG. When retrieval system provide misleading content, standard RAG would overly rely on the document and generate answer that are faithful to the document while not factually true. Our xRAG model opt to rely on the internal

knowledge of LLM and being robust to the misleading content. In Appendix H, we include more cases about xRAG including several error analysis.

# 7 Conclusion

In this work, we present xRAG, an innovative context compression method tailored for retrieval-augmented generation. For knowledge-intensive tasks, xRAG can be significantly faster than RAG while maintaining comparable performance. We are excited about the future of this modality-based retrieval-augmented system and plan to further improve its performance in the areas of reasoning over embedding, handling multiple documents, and combining with multi-vector retrieval.

# 8 Acknowledgement

We would like to express our sincere gratitude to the anonymous reviewers for their thorough review, insightful comments, and constructive suggestions, which have significantly improved the quality of this manuscript. This work paper is supported (in part) by the State Key Laboratory of General Artificial Intelligence.

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

# A  Comparison between different Context Compression Models

In Table 5, we present a detailed comparison of various context compression models, emphasizing their real-world applicability. This comparison focuses on two key aspects: (1) Plug-and-play capability, which assesses whether dataset-specific tuning is necessary for new, unseen data; (2) Memory efficiency, which evaluates if additional memory space is required to store the compressed information, such as high-dimensional vectors typically used in soft prompting methods.

| Model | Specifically designed for RAG | Maximun Compression Rate | Approach | Plug-and-Play | Memory Efficient |
|---|---|---|---|---|---|
| AutoCompressor [15] | ✗ | x15 | Soft Prompting | ✗ | ✗ |
| Gist [59] | ✗ | x26 | Soft Prompting | ✗ | ✗ |
| ICAE [20] | ✗ | x8 | Soft Prompting | ✓ | ✗ |
| LLMLingua [29] | ✗ | x20 | Prompt Editing | ✓ | ✓ |
| Selective Context [46] | ✗ | x5 | Prompt Editing | ✓ | ✓ |
| Token Elimination [85] | ✓ | x10 | Attention Filtering | ✓ | ✓ |
| FilCo [74] | ✓ | x2 | Prompt Editing | ✗ | ✓ |
| RECOMP [79] | ✓ | x16.6 | Prompt Editing | ✓ | ✓ |
| xRAG | ✓ | x178 | Modality Fusion | ✓ | ✓ |

Table 5: Comparison between different compression methods from their setting to design principle.

# B  Instructions for Paraphrase Pretraining

The list of instructions for Paraphrase Pretraining is shown in Table 6. They present the same meaning with natural language variance.

- "Background: [X] means the same as [D] "
- "Background: [X] Can you put the above sentences in your own terms? [D] "
- "[X] Please provide a reinterpretation of the preceding background text. [D] "
- "These two expressions are equivalent in essence:(1) [X] (2) [D] "
- "Background: [X] is a paraphrase of what? [D] "
- "[X] Could you give me a different version of the background sentences above? [D] "
- "In other words, background: [X] is just another way of saying: [D] "
- "You're getting across the same point whether you say background: [X] or [D] "
- "[X] After uppacking the ideas in the background information above, we got: [D] "
- "[X] Please offer a restatement of the background sentences I've just read. [D] "
- "Background: [X] , which also means: [D] "
- "Strip away the mystery, and you'll find [X] is simply another rendition of: [D] "
- "The essence of background: [X] is captured again in the following statement: [D] "

Table 6: Instructions used for Paraphrase Pretraining where [X] and [D] are placeholders for projected retrieval feature $\mathbf{W}(E)$ and document D.

# C   Details for Instruction Tuning Dataset

After collecting the raw data from different categories, we use templates[4] from FLAN [77] to construct instruction tuning dataset. In Table 7 we list an overview and in Table 8 we list the detailed information for each subtask of our dataset. For QA tasks that lack an explicit context, we perform a retrieval operation within the corpus $\mathbb{D}$ to identify the most relevant document to serve as context. This approach is akin to the retrieval-augmented instruction tuning depicted in [55, 51].

| Task Type | # Involved datasets | # Train | # Prompt | # Label |
|---|---|---|---|---|
| Reading Comprehension | 7 | 488,344 | 447.62 | 30.34 |
| Summarization | 3 | 81,821 | 483.49 | 53.29 |
| Open Domain QA | 7 | 385,173 | 203.55 | 20.09 |

Table 7: Overall statistics of Instruction Tuning dataset.

| Task Type | Dataset | # Train | # Prompt Len | # Label Len |
|---|---|---|---|---|
| Reading Comprehension | CoQA [66] | 7101 | 617.98 | 77.75 |
| | DROP [17] | 76098 | 356.06 | 3.86 |
| | NarrativeQA [40] | 32747 | 702.39 | 7.86 |
| | PubMedQA [32] | 1000 | 397.91 | 65.4 |
| | QuAIL [67] | 10246 | 512.9 | 2.0 |
| | SQuAD v2 [65] | 130319 | 214.54 | 6.87 |
| | PwC [20] | 241564 | 571.35 | 53.07 |
| Open Domain QA | NQ [41] | 87925 | 203.62 | 5.976 |
| | TriviaQA [33] | 78785 | 216.1 | 6.49 |
| | CommonsenseQA [72] | 9741 | 223.64 | 2.0 |
| | WikiQA [80] | 1040 | 192.89 | 40.79 |
| | YahooQA[5] | 87358 | 196.56 | 56.7 |
| | FreebaseQA [31] | 20353 | 218.49 | 4.87 |
| | MSMarco [6] | 99994 | 194.82 | 15.91 |
| Summarization | CNN/DM [69] | 100000 | 616.99 | 63.37 |
| | SamSum [21] | 14731 | 187.87 | 29.12 |
| | DialogSum [11] | 12460 | 247 | 37.61 |

Table 8: Detailed data statistics for our Context-aware Instruction Tuning Dataset.

---

[4]https://github.com/google-research/FLAN/blob/main/flan/templates.py

# D Implementation Details

For the language models we use, Mixtral-8x7b is approximately 6.5 times larger in scale compared to Mistral-7b and features a divergent architectural approach—specifically, a dense versus mixture-of-experts design. For our assessments, we employed the instruction-tuned variants of these models.

Owing to efficiency constraints, we opted not to perform on-the-fly retrieval. Instead, we pre-constructed a retrieval index using the efficient and robust multi-vector retriever, ColBERT-v2 [68], from which we retrieved the *top-1* ranked document for inclusion in our instruction-tuning dataset and for the evaluation of downstream tasks. Subsequently, we re-encoded these documents using the embedding model of interest (e.g., SFR). This strategy allows us to iterate data-centric experiments quickly. All experiments are conducted on the a setup of 8xNvidia A100 GPUs.

In Table 9 and Table 10, we list the hyperparameters for Paraphrase Pretraining and Context-aware Instruction Tuning.

| Hyperparameter | Assignment |
| --- | --- |
| optimizer | AdamW |
| learning rate | 6e-3 |
| lr scheduler type | linear |
| warmup ratio | 0.03 |
| weight dacay | 0.0 |
| epochs | 1 |
| flash attention | True |
| batch size | 12 |
| gradient accumulation steps | 4 |
| num GPUs | 8 |
| max sequence length | 336 |
| max train samples | 2,000,000 |

Table 9: Hyperparameters for Paraphrase Pretraining.

| Hyperparameter | Assignment |
| --- | --- |
| optimizer | AdamW |
| learning rate | 2e-5 |
| lr scheduler type | linear |
| warmup ratio | 0.03 |
| weight dacay | 0.0 |
| epochs | 1 |
| KL $\alpha$ | 2.0 |
| KL temperature | 1.0 |
| flash attention | True |
| batch size | 4 |
| gradient accumulation steps | 2 |
| num GPUs | 8 |
| max sequence length | 1024 |
| max train samples | 955,338 |

Table 10: Hyperparameters for Context-aware Instruction Tuning.

# E About different Embedding Models

In our primary experiments, we use the SFR model as our default sentence embedding model. This section delves into the effects of different embedding models. We examine four universal text embedding models: E5-Mistral and E5-Large [73] alongside BGE-Large and BGE-Base [78]. Additionally, we assess two retrieval-specific models: Dragon [49] and DPR [35]. The configurations of the different retrievers and their MTEB scores[6]—a metric indicating their general sentence representation capability—are listed in Table 11. To isolate the impact of potentially different retrieved documents, we ensure that all models utilize the same *top-1* document. The performance is averaged over four question answering datasets. A general pattern is that embedding models with stronger sentence representation capabilities tend to further enhance the downstream performance. Remarkably, the Dragon model, despite being a BERT-base-sized retrieval-specific model, outperforms general text embedding models that are twice its size (BGE-Large).

| Model | | Model Size | Embedding Dim | Universal Embedding | MTEB Score | Average | | |
|---|---|---|---|---|---|---|---|---|
| | | | | | | Performance | Resilience | Boost |
| **Mistral-7b** | | | | | | | | |
| | w/o retrieval | | | | | 37.2 | - | - |
| | w retrieval | | | | | **46.3** | 74.1% | **29.5%** |
| xRAG | w SFR | 7B | 4096 | ✓ | 67.56 | 44.5 | 82.3% | 22.2% |
| | w E5-Mistral | 7B | 4096 | ✓ | 66.63 | 44.0 | 84.0% | 20.6% |
| | w E5-Large | 335M | 1024 | ✓ | 62.25 | 42.1 | 80.2% | 19.6% |
| | w BGE-Large | 335M | 1024 | ✓ | 64.23 | 41.6 | 78.2% | 19.8% |
| | w BGE-Base | 109M | 768 | ✓ | 63.55 | 41.2 | 78.7% | 18.9% |
| | w Dragon | 109M | 768 | ✗ | - | 42.1 | **84.2%** | 16.9% |
| | w DPR | 109M | 768 | ✗ | - | 40.5 | 77.4% | 18.2% |

Table 11: Ablation on different sentence embedding models.

---

[6] https://huggingface.co/spaces/mteb/leaderboard

# F   Analysis on Mistral-7b

In Figure 6, we list the Resilience rate and Boost Rate on Mistral-7b model, which exhibit same pattern with Mixtral-8x7b model.

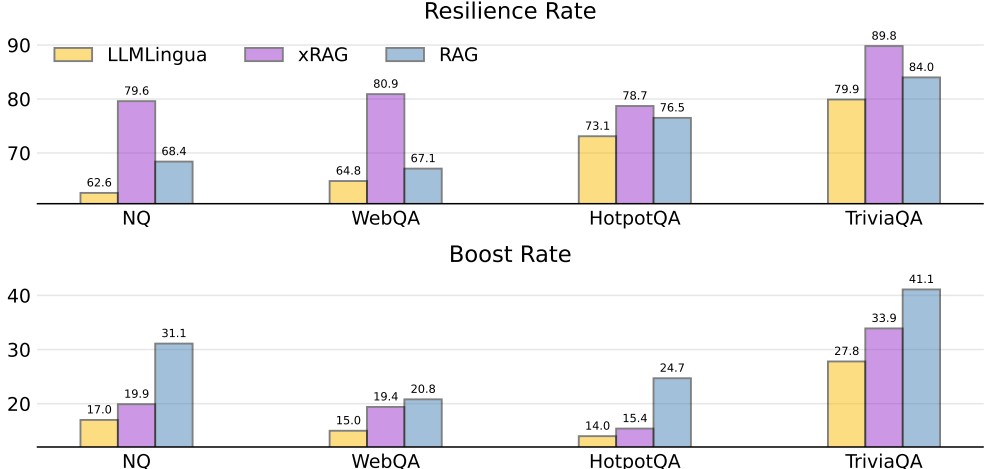

Figure 6: Robustness and effectiveness analysis on 4 QA datasets with Mistral-7b model.

# G  Limitations

We discuss the limitations of our framework as follows:

- In this work, we only consider the most commonly used retrieval system—single dense vector retrieval, while sparse retrieval methods such as BM25 or multi-vector retrieval methods like ColBERT are not included. We believe that combining these methods would be a promising direction for xRAG, as sparse vectors could complement dense vectors, and multi-vector retrieval would provide xRAG with more flexibility by not condensing all information into one token.

- Currently, xRAG delivers decent performance when a relevant document is fetched; however, it lags behind RAG by a considerable margin in tasks that require reasoning (such as HotpotQA and FactKG). One possible reason is that during the training phase of xRAG, reasoning-relevant data is not provided. How to make xRAG a better reasoner remains our future work.

- We only consider the *Top-1* retrieval setting, while ensembling multiple relevant documents has been shown to be effective for RAG systems due to the complementary information contained in *Top-K* documents. We believe there is potential advantage for xRAG to scale to multi-document settings, as the input length of xRAG for multi-documents scales by a factor of 1, while for RAG, it scales by the document length factor.

## H  More interesting cases

- In Figure 7, we report a failure case of xRAG. In this case, retrieval alone is not enough to derive the final answer and the LLM is required to perform reasoning over retrieved document (the listed universities are all located in Switzerland).

- An interesting example is shown in Figure 8, when the retrieved document is a list of a characters in the book Discworld, the RAG model would respond with a fictional character, while xRAG generate the right answer by focusing on the relevant part of the document.

- In Figure 9, even when the retrieved document is relevant, RAG would still hallucinate while xRAG could generate the right answer based on the document.

- In Figure 10, the retriever mistakenly fetch the wrong document (Phantom of the Opera of interest is a music rather than a file) and RAG would be misled while xRAG remain robust to generate the correct answer.

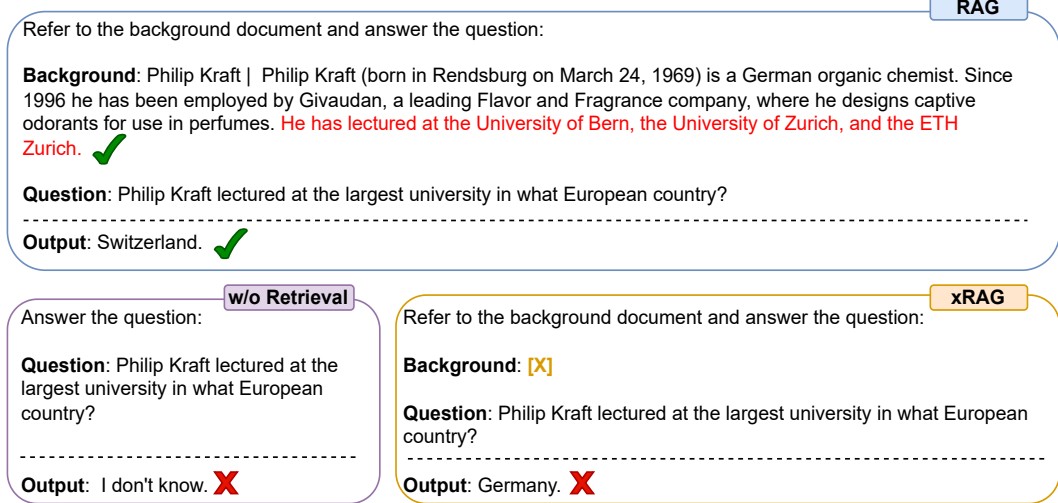

Figure 7: Failure case of xRAG when reasoning is required to derive the final answer.

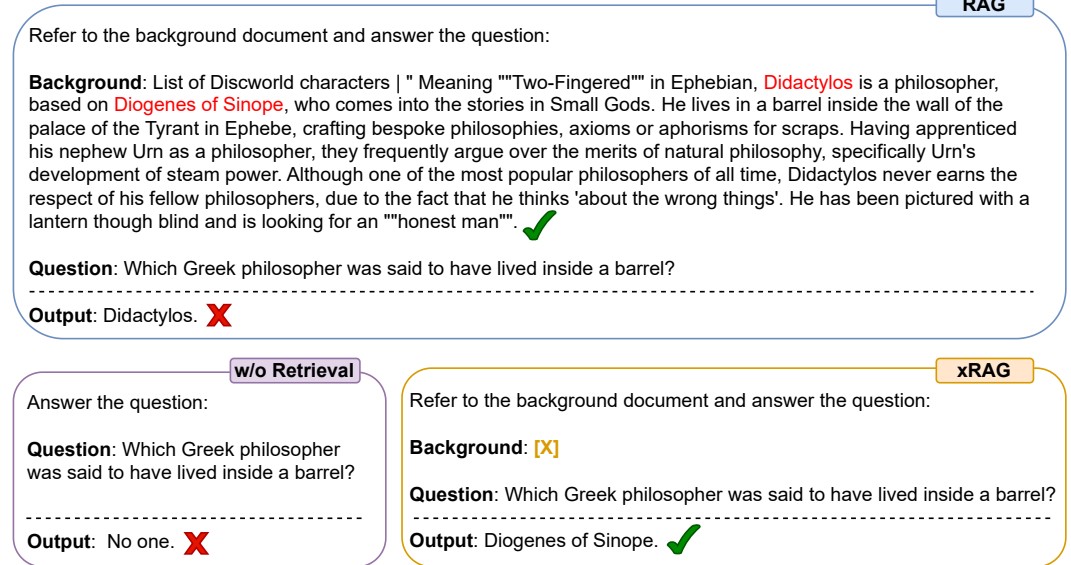

Figure 8: xRAG correctly locates the relevant part in a long document by selecting Diogenes of Sinope as the answer rather than Didactylos, a fictional character in the book Discworld.

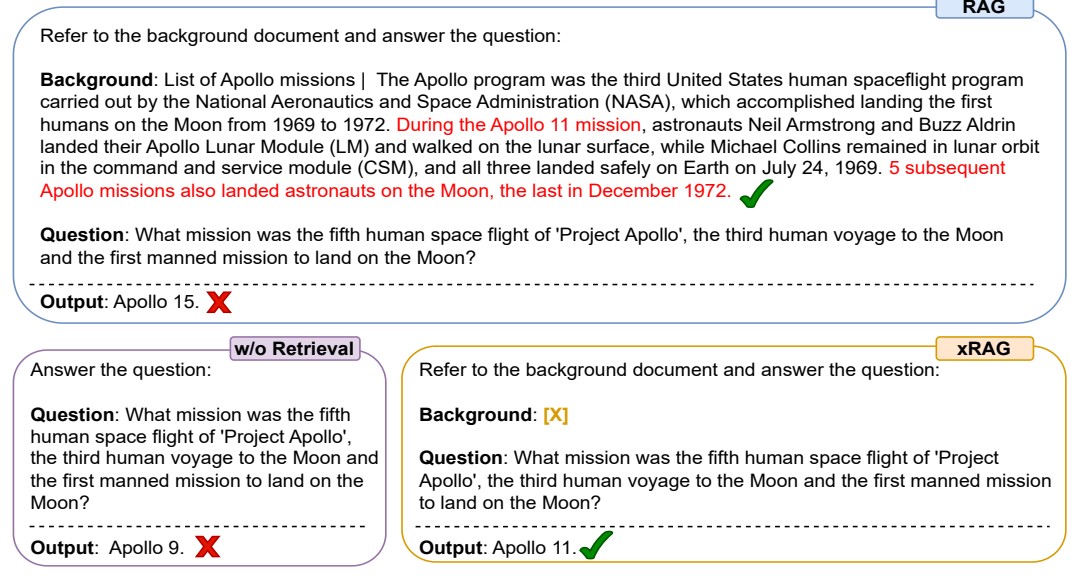

Figure 9: xRAG correctly locates the relevant part in a long document while RAG would still hallucinate the wrong answer.

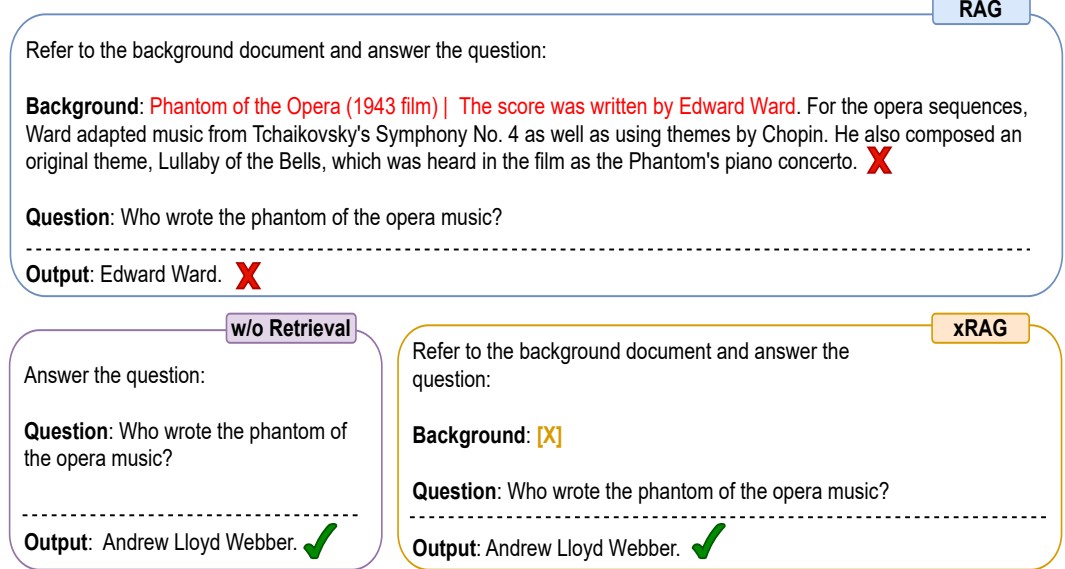

Figure 10: xRAG correctly locates the relevant part in a long document while RAG would still hallucinate the wrong answer.

