# OpenReview forum: "xRAG: Extreme Context Compression for Retrieval-augmented Generation with One Token"
_NeurIPS.cc/2024/Conference — NeurIPS 2024 poster_

### Official Review · Reviewer_2bxX · 2024-07-07

**Soundness:** 2
**Presentation:** 3
**Contribution:** 2
**Rating:** 4
**Confidence:** 5

**Summary:**

This paper proposes xRAG, a method to map the retriever's embeddings into LM's representation space with one token. While both the retriever and LM are fixed, xRAG trains a simple projector to adapt to them by paraphrase pretraining and context-aware instruction tuning. xRAG can significantly outperform non-retrieval and previous context compression methods and matches the performance of the uncompressed one, saving the RAG FLOPs by more than 3 times.

**Strengths:**

- The compression rate from the long context to one token is high
- The only trainable component constitutes less than 0.1% of the LLM’s parameters, which can serve as an efficient plug-in
- xRAG can match or surpass the original RAG with only one token for one document, and the analysis of the Resilience Rate and the Boost Rate is reasonable and interesting
- In general, the paper is well written

**Weaknesses:**

- The main experiment in this paper is conducted only on one retrieved document, whose average length is just 175, but I think this is a **very short context** for modern LLMs (and thus the inference time is not a bottleneck). I think proving the effectiveness of xRAG on a real-long context (more than thousands of tokens) of multiple documents is very important to see this method's real-world value

- It is essential to consider the total training FLOPs of the projector. Although we don't need the actual optimization of LM and retriever, we also need the forward process of these models in the projector's training so that **the cost can be significant**, especially when the amount of total two-stage training samples is very, very large (near 3M)

- The projector is **tightly coupled** with the retriever and LM. It is hard to prove the generalization abilities of xRAG to other black-box production LLMs where the projector cannot be trained directly from the LM's feedback, which may limit its potential impact

- In Section 4.3, the requirements of the baselines include **no need for dataset-specific tuning**. However, how to explain that NQ, TriviaQA (two evaluation tasks) are included in the Context-aware Instruction Tuning data?

- Missing baselines of context compression methods other than LLMLingua, like Gist-COCO. Will adding **more compression tokens** benefit the performance?

[1]: Li, Xinze, et al. "Say more with less: Understanding prompt learning behaviors through gist compression." arXiv preprint arXiv:2402.16058 (2024).

**Questions:**

- For plugging in multiple retrieved documents, should we devise another form of training task or simply adopt the same projector multiple times? This is an interesting discussion that I would like to see
- How simple can the projector be? I would like to see the performance of different design choices of the projector, such as Q-Former mentioned in the paper
- What is the data source for Paraphrase Pretraining? I cannot find it in the paper

**Limitations:**

Yes.

---

> ### Author Rebuttal · Authors · 2024-08-06
>
> Dear Reviewer,
>
> We greatly appreciate your time and effort in reviewing our paper. Here, we would like to address your concerns point by point:
>
> - **About multiple documents:**
>
> Please refer to our general response about multiple documents in xRAG.
>
> - **The cost of training**
>
> The training cost of xRAG is minimal compared to full parameter tuning since the only trainable component is a two-layer MLPs. In our setup (8xA00), full parameter tuning of a 7B dense model with 1 million samples takes about 24 hours. In the same amount of time, xRAG (8x7b) can finish training on 2.9 million samples (paraphrase pre-training + context-aware instruction tuning). Moreover, by caching all the document embeddings, we can further speed up the training by 25%. We appreciate your suggestion and will include this information in the revised version of our paper.
>
> - **xRAG for black-box LLM**
>
> From the user side, it is true that xRAG can only be applied to models with open weights. However, with the increasing popularity of cloud vector databases, LLM API providers could still benefit from xRAG if users can provide the index of retrieved documents in the vector database.
>
> - **About dataset-specific tuning.**
>
> When we refer to "without dataset-specific tuning," we mean that we adopt a one-for-all evaluation setting where the evaluation results from various datasets come from the same model, as opposed to one-for-one evaluation which requires dataset-specific tuning. To exclude the impact of the corresponding training set (NQ and TQA), we refer to Table 4 of our paper, where similar results are obtained even with only reading comprehension data involved (NQ and TQA excluded). Additionally, the LoRA results in Table 3 confirm that the improvement of xRAG does not stem from task knowledge inherited during context-aware instruction tuning.
>
> - **Missing baselines of context compression methods other than LLMLingua, like Gist-COCO. Will adding more compression tokens benefit the performance?**
>
> Thank you for pointing out this missing reference. Similar to Gist Token [1], Gist-COCO relies on gist representation, which might not be applicable in real-world RAG settings involving millions of documents. We will add this paper to the final version of xRAG in both the Related Work section and Appendix A. Regarding the addition of more compression tokens, we believe it is a balance between efficiency and efficacy. In Appendix G, we discuss that xRAG could be combined with a multi-vector or Matryoshka embedding system where the compression ratio is configurable. We believe this is an exciting direction for xRAG.
>
> - **For plugging in multiple retrieved documents, should we devise another form of training task or simply adopt the same projector multiple times? This is an interesting discussion that I would like to see**
>
> Please refer to our general response about multiple documents in xRAG.
>
> - **How simple can the projector be? I would like to see the performance of different design choices of the projector, such as Q-Former mentioned in the paper**
>
> As discussed in Section 3.4, the key design principle is the simplicity of the system. To our knowledge, the simplest form for a projector is an MLP architecture. We leave the exploration of more advanced and complex architectures for future work.
>
> - **What is the data source for Paraphrase Pretraining? I cannot find it in the paper**
>
> Apology for the confusion. The data source for Paraphrase Pretraining is the same Wikipedia dump used as the retrieval corpus. We will make this clear in the revised version of our paper.
>
> [1] Mu, Jesse, Xiang Lisa Li and Noah D. Goodman. “Learning to Compress Prompts with Gist Tokens.” *ArXiv* abs/2304.08467 (2023): n. pag.

---

> ### Comment · Reviewer_2bxX · 2024-08-11
>
> Thanks to the authors. I have read the rebuttal carefully. I decided to not change my scores since I feel the limitations of this work still outweigh reasons to accept. (1): The additional training cost seems not trivial; (2): The efficiency gain in super-long context (not only just 3 docs) is not included.

---

> > ### Author Response · Authors · 2024-08-11
> >
> > Dear Reviewer,
> >
> > We sincerely appreciate your thoughtful feedback and welcome the opportunity to address the points you've raised.
> >
> > - **Concern: The additional training cost seems non-trivial**
> >
> > Similar to how LLaMA[1] "violates" the scaling law by prioritizing the inference budget, we believe that inference cost is of utmost importance for a context compression method, especially in real-world RAG scenarios involving millions of documents.  This principle guided xRAG's design: by reusing existing document embeddings and introducing only a two-layer MLP into the existing LLM, we've minimized additional complexity.
> >
> > While our method prioritizes inference-time efficiency, it's worth noting that **the training cost is also minimal compared to existing compression methods such as ICAE[2] and LLMLingua[3].** The only trainable component in xRAG is the newly introduced MLP. We would greatly appreciate if you could point us towards any references to context compression methods with trivial training costs that we may have overlooked.
> >
> > ---
> >
> > - **Concern: The efficiency gain in handling super-long contexts (beyond just 3 documents) is not included.**
> >
> > RAG is widely recognized as a technique designed to alleviate the need for LLMs to process long contexts by chunking documents into smaller pieces and retrieving the most relevant ones. This is the typical operational model of modern RAG systems. As such, long-context processing and RAG represent two distinct paradigms for LLMs.
> >
> > Research has shown that useful information generally appears within the top-k documents, and RAG performance tends to plateau as more documents are involved [4] [5]. Consequently, handling super-long contexts falls outside the scope of RAG and, by extension, beyond the scope of xRAG.
> >
> > However, to address your concern about efficiency with more documents, we've conducted additional tests. If the "super-long" context you mentioned falls within the top-k chunks of RAG (where k is typically less than 10), here are the efficiency results for top-10 documents, following the benchmark setting outlined in Section 5.2 of our paper:
> >
> >
> >
> > | Top-10 chunks | CUDA Time (s) | GFLOPs         | Peak Mem (GiB) |
> > | ------------- | ------------- | -------------- | -------------- |
> > | RAG           | 3.58s         | 10712.22       | 20.42          |
> > | xRAG          | 0.62s (x5.7)  | 529.33 (x20.2) | 13.84 (x1.4)   |
> >
> > As demonstrated, xRAG maintains significant efficiency gains even when processing a larger number of documents.
> >
> > We appreciate your insightful feedback and look forward to further discussion on these points.
> >
> > ---
> > [1] Touvron, Hugo, Thibaut Lavril, Gautier Izacard, Xavier Martinet, Marie-Anne Lachaux, Timothée Lacroix, Baptiste Rozière, Naman Goyal, Eric Hambro, Faisal Azhar, Aurelien Rodriguez, Armand Joulin, Edouard Grave and Guillaume Lample. “LLaMA: Open and Efficient Foundation Language Models.” ArXiv abs/2302.13971 (2023): n. pag.
> >
> > [2] Ge, Tao, Jing Hu, Xun Wang, Si-Qing Chen and Furu Wei. “In-context Autoencoder for Context Compression in a Large Language Model.” ArXiv abs/2307.06945 (2023): n. pag.
> >
> > [3] Jiang, Huiqiang, Qianhui Wu, Chin-Yew Lin, Yuqing Yang and Lili Qiu. “LLMLingua: Compressing Prompts for Accelerated Inference of Large Language Models.” Conference on Empirical Methods in Natural Language Processing (2023).
> >
> > [4] Lewis, Patrick, Ethan Perez, Aleksandara Piktus, Fabio Petroni, Vladimir Karpukhin, Naman Goyal, Heinrich Kuttler, Mike Lewis, Wen-tau Yih, Tim Rocktäschel, Sebastian Riedel and Douwe Kiela. “Retrieval-Augmented Generation for Knowledge-Intensive NLP Tasks.” ArXiv abs/2005.11401 (2020): n. pag.
> >
> > [5] Shi, Weijia, Sewon Min, Michihiro Yasunaga, Minjoon Seo, Rich James, Mike Lewis, Luke Zettlemoyer and Wen-tau Yih. “REPLUG: Retrieval-Augmented Black-Box Language Models.” ArXiv abs/2301.12652 (2023): n. pag.

---

> > > ### Comment · Reviewer_2bxX · 2024-08-12
> > >
> > > Thanks for your response. How about xRAG's performance compared with RAG (no compression) in top-3 and top-10 chunks? A partial evaluation should be fine. I will increase my score if this question is clear.

---

> > > > ### Author Response · Authors · 2024-08-12
> > > >
> > > > Dear Reviewer,
> > > >
> > > > Thank you for your continued engagement and insightful question. We appreciate the opportunity to provide further clarification.
> > > >
> > > > As requested, we have conducted additional experiments comparing xRAG and RAG performance with multiple chunks. The results are as follows:
> > > >
> > > > |           | NQ   | TriviaQA | WebQ | HotpotQA | Doc Length|
> > > > | --------- | ---- | -------- | ---- | -------- | ----- |
> > > > | xRAG-top3 | 41.4 | 67.3     | 41.1 | 35.3     | 3     |
> > > > | RAG-top3  | 45.6 | 69.1     | 40.4 | 40.2     | 525.6 |
> > > >
> > > > We want to emphasize that this version of xRAG, as described in our general response, **serves as a feasibility test** rather than a rigorous evaluation of xRAG's full potential with multiple chunks. This implementation modifies only a subset of data (summarization) in our context-aware instruction tuning while maintaining the original model architecture, paraphrase pretraining stage, and objective function. Given that the summarization data is typically divided into at most 3 chunks (with a maximum length of 512 tokens per sample and 180 tokens per chunk), an evaluation with top-10 chunks is beyond the scope of the current xRAG implementation.
> > > >
> > > > While xRAG currently falls short of RAG in this multi-chunk setting, we insist **this does not diminish our core contribution**. The expansion to multiple chunks represents an incremental improvement (1 to N) of the existing framework. While valuable, we consider it less fundamental compared to our primary  0 to 1 innovation: efficient RAG with modality fusion.
> > > >
> > > > We appreciate your thoughtful consideration and hope this additional information addresses your query satisfactorily. We remain committed to advancing this promising research direction and value the constructive feedback from the review process.
> > > >
> > > >
> > > > Best regards,
> > > >
> > > > Authors

---

### Official Review · Reviewer_MKxH · 2024-07-11

**Soundness:** 2
**Presentation:** 3
**Contribution:** 2
**Rating:** 3
**Confidence:** 5

**Summary:**

The paper on xRAG presents an innovative approach to context compression in retrieval-augmented generation, achieving significant efficiency gains while maintaining performance. The method's compatibility with various language models and preservation of the plug-and-play nature are notable strengths. However, the added complexity and dependency on high-quality embeddings might pose challenges. The whole paper is clear and easy to understand.

**Strengths:**

1. This paper introduces xRAG, a novel context compression method that effectively merges document embeddings into the language model’s representation space through modality fusion, which is an attractive topic.
2. The xRAG achieves significant context compression compared with baselines.
3. The plug-and-play nature makes it easy to be applied with other backbone models.

**Weaknesses:**

1. The introduction of a modality bridge and the requirement for a pretrained modality encoder adds complexity to the system.
2. Will the bias in projector tuning cause hallucinations?
3. The performance of xRAG is likely highly dependent on the quality of the dense document embeddings.
4. The compression method, while efficient, might result in the loss of information present in the original documents. Do we need such extreme compression since most of the queries are not too long? Or how to balance the compressed tokens with the performance?
5. The concepts are so confused. Sometimes, the author uses the "modality bridge" and also uses the "projector". I think they are the same module right?
6. The paper mentions using a two-layer MLP as the projector but doesn't explore how different projector architectures might impact performance.
7. The paper suggests that xRAG is more robust than RAG when dealing with irrelevant documents but doesn't provide a clear mechanism for how xRAG achieves this robustness.
8. How will the hyper parameter alpha influence the final results?
9. How to ensure the compressing process filters the irrelevant information rather than the crucial ones?
10. Will sensitive information like numbers or times be kept by the projector after compressing?

**Questions:**

See weakness.

**Limitations:**

Yes

---

> ### Author Rebuttal · Authors · 2024-08-06
>
> Dear Reviewer,
>
> Appreciate your time and effort in reviewing our paper. We would like to address your concerns point by point:
>
> - **Added complexity to the system**
>
> In xRAG, we have emphasized throughout our paper that our key design principle is to add minimal complexity to the existing RAG system. In Section 3.4, we explicitly state that our approach **does not require an additional pretrained modality encoder** but instead utilizes existing document embeddings from the retrieval datastore. This design choice ensures **zero** additional space complexity, enabling practical scalability in real-world RAG systems handling millions of documents. To make our method plug-and-play, we freeze the LLM and retriever, and the only trainable component is the projector.
>
> - **Will the Bias in Projector Tuning Cause Hallucinations?**
>
> In Section 6.1, we introduce a metric called Resilience Rate, which measures how retrieval (RAG or xRAG) affects LLM performance and prevents hallucinations. Empirical results show that our method is more robust and produces fewer hallucinations than vanilla RAG.
>
> - **Dependence on the Quality of Dense Document Embeddings**
>
> It is widely acknowledged that the performance of the modality encoder heavily impacts downstream performance, and xRAG is no exception. However, xRAG is not limited to only strong embedding models. In Section 6.2, we demonstrate the effectiveness of xRAG across various embedding models, showing that even with a four-year-old DPR model, xRAG still yields over 8% improvement.
>
> - **The compression method, while efficient, might result in the loss of information present in the original documents. Do we need such extreme compression since most of the queries are not too long? Or how to balance the compressed tokens with the performance?**
>
> According to Claude Shannon's information theory, one classic estimate of the amount of information per English word is 11.82 bits per word [1]. This means, in theory, we have enough raw state to encode more tokens with a single embedding without loss of information. In that sense, we can be more efficient when dealing with real-world multi-round, multi-document RAG settings, where prompt length could be up to thousands of tokens. More importantly, in  Section 5.2, we have demonstrated that the current xRAG could bring considerable efficiency improvement even on a high-end server GPU.
>
> As for balancing compression and performance, we discuss in Appendix G that xRAG can be combined with a multi-vector or Matryoshka embedding system where the compression ratio is configurable. We believe this is an exciting direction for xRAG.
>
> - **The concepts are so confused. Sometimes, the author uses the "modality bridge" and also uses the "projector". I think they are the same module right?**
>
> They refer to the same module that connects two modalities.
>
> - **The paper mentions using a two-layer MLP as the projector but doesn't explore how different projector architectures might impact performance.**
>
> As discussed in Section 3.4, the key design principle is simplicity. Therefore, we chose the simplest and most commonly used projector: a two-layer MLPs. Exploration of more advanced architectures is left for future work.
>
> - **The paper suggests that xRAG is more robust than RAG when dealing with irrelevant documents but doesn't provide a clear mechanism for how xRAG achieves this robustness.**
>
> One reason for xRAG's robustness is that it is trained with retrieved context in the second training phase, which improves the LLM's resilience to noisy retrievals. This phenomenon is also observed in [2][3]. Additionally, xRAG can avoid word-by-word repetition seen in RAG when dealing with noisy retrieval results.
>
> - **How will the hyper parameter alpha influence the final results?**
>
> We tested several values for the hyperparameter alpha ({0.1, 0.5, 1.0, 2.0, 3.0}) and found that 2.0 gave the best results on the validation set. No significant differences were observed when alpha ≥1.
>
> - **How to ensure the compressing process filters the irrelevant information rather than the crucial ones?**
>
> Our paper, like previous context compression papers, does not claim to achieve lossless compression. It is a balance between efficiency and efficacy. We do not filter out any information; instead, we aim to recover the original information with a single embedding vector. Our novel training paradigm, from paraphrase pretraining to context-aware instruction tuning, is designed to achieve this goal.
>
> - **Will sensitive information like numbers or times be kept by the projector after compressing?**
>
> xRAG has the ability to retain sensitive information such as numbers and times. As shown in Figure 9 of our paper, xRAG successfully identifies important numbers and answers questions correctly.
>
> Thank you again for your review. We hope that our response alleviates some of your concerns and that you might consider raising the score of our paper.
>
> Best regards,
>
> Authors
>
> [1] Claude Elwood Shannon. “Prediction and Entropy of Printed English.” (1951).
>
> [2]  Lin, Xi Victoria, Xilun Chen, Mingda Chen, Weijia Shi, Maria Lomeli, Rich James, Pedro Rodriguez, Jacob Kahn, Gergely Szilvasy, Mike Lewis, Luke S. Zettlemoyer and Scott Yih. “RA-DIT: Retrieval-Augmented Dual Instruction Tuning.” *ArXiv* abs/2310.01352 (2023): n. pag.
>
> [3] Luo, Hongyin, Yung-Sung Chuang, Yuan Gong, Tianhua Zhang, Yoon Kim, Xixin Wu, Danny Fox, Helen M. Meng and James R. Glass. “SAIL: Search-Augmented Instruction Learning.” *Conference on Empirical Methods in Natural Language Processing* (2023).

---

> > ### Comment · Reviewer_MKxH · 2024-08-13
> >
> > Thanks to the author for the responses and some of my concerns have been solved. The necessity of compressing the context into one token remains questionable and I will keep my score.

---

> > > ### Author Response · Authors · 2024-08-13
> > >
> > > Dear Reviewer,
> > >
> > > Thank you for taking the time to engage in further discussion and for acknowledging that some of your concerns have been addressed. We would like to clarify the rationale behind compressing the context into a single token.
> > >
> > > As demonstrated in our paper, our approach **not only achieves a higher compression rate but also results in better accuracy compared to existing context compression methods.** We believe that improving both performance and efficiency is crucial in advancing the state-of-the-art.
> > >
> > > Could you please elaborate on why a context compression method with better performance and efficiency is not preferred?
> > >
> > > We appreciate your insights and look forward to your feedback.
> > >
> > > Sincerely,
> > >
> > > Authors

---

### Official Review · Reviewer_jbwR · 2024-07-11

**Soundness:** 2
**Presentation:** 3
**Contribution:** 2
**Rating:** 4
**Confidence:** 4

**Summary:**

This paper proposes xRAG, a context compression method designed specifically for retrieval-augmented generation. xRAG redefines the use of document embeddings in dense retrieval by integrating them as features from the retrieval modality.  It achieves an extreme compression rate to only one token. The authors perform experiments to  demonstrate the effectiveness of the proposed method.

**Strengths:**

1. Using modality fusion bridge to connect the embeddings from dense retrieval and LLMs for retrieval-augmented generation.

2. Extremely compress the input tokens from RAG to only one token.

3. Comparable performance with lower FLOPs.

**Weaknesses:**

1. The proposed method relies heavily on the performance of dense retrieval models. The dense retrieval models have poor generalization for out-of-domain [1], which does not match the general ability of LLMs, but the experiment in this paper is only conducted on wiki-based knowledge bases of Q&A tasks.

2. The specific details of selection and training method of dense retrieval model in xRAG are needed.

3. How xRAG performs on long-context RAG? Can only one token represents the semantics of the retrieved long-documents?


[1] Back to Basics: A Simple Recipe for Improving Out-of-Domain Retrieval in Dense Encoders

**Questions:**

1. How xRAG performs on long-context RAG? Can only one token represents the semantics of the retrieved long-documents?

2. How xRAG selects or trains the dense retrieval model?

3 How xRAG performs on the domain that shifts from the training domain of dense retrieval model?

**Limitations:**

Authors have discussed the limitations of this paper in Section G of Appendix.

---

> ### Author Rebuttal · Authors · 2024-08-06
>
> Dear Reviewer,
>
> We greatly appreciate your time and effort in reviewing our paper. Here, we would like to address your concerns point by point:
>
> - **Out-of-Domain Generalization**
>
> Thank you for bringing up this issue. If we understand your concerns correctly, they can be divided into two parts:
>
> 1. The Out-of-Domain Generalization of Dense Retrieval Models: We acknowledge that dense retrieval models face challenges with cross-domain generalization, yet they remain the de facto approach for modern RAG systems [1][2][3]. Importantly, dense retrieval models (such as DRAGON[4] and ColBERT[5]) have demonstrated stronger generalization compared to their sparse counterparts (such as BM25), as evidenced by benchmarks like BEIR[6] and MTEB[7]. These models are all compatible with xRAG.
>
> 2. The Out-of-Domain Generalization of xRAG: The ultimate goal of xRAG is to create compatible representations between dense embeddings and their textual form within the LLM representation space using a modality fusion approach. We believe a well-trained projector can transform any text into the embedding space. To verify this, we tested our xRAG model (paraphrase pre-trained in the Wikipedia) in the biomedical domain. Specifically, we tested on PubMedQA[8], BioASQ-Y/N[9], and ChemProt[10] using PubMed[11] as our retrieval corpus to evaluate the cross-domain generalization ability of xRAG. By comparing the first two rows in the table below, we observed that simply adding one document token, xRAG gives considerable improvement over the Mistral-7b baseline, demonstrating the cross-domain generalization of xRAG. Moreover, xRAG can easily adapt to other domains because the first stage of training pipeline: paraphrase pretraining, is a self-supervised process that does not require handcrafted labels. We sample documents from PubMed and pre-train xRAG on it, which further improve the performance as shown in the last row.
>
> | Model | PubMedQA | BioASQ-Y/N | ChemProt |
> | --- | --- | --- | --- |
> | Mistral-7b | 43.6 | 74.5 | 66.7 |
> | xRAG-7b | 49.2 | 82.1 | 71.8 |
> | xRAG-7b (PubMed) | 51.1 | 83.2 | 72.8 |
>
> - **Details about Training and Selection of Dense Models:**
>
>     As stated in Section 4.2 of our paper, we selected SFR[12] as our dense embedding model, which, at the time of writing, held the leading position on the MTEB leaderboard. We did not train this model ourselves, allowing us to reuse the offline constructed embeddings and adding ZERO overhead to the existing RAG system, the core of xRAG as detailed in Section 3.4.
>
> - **Long-Context RAG:**
>
>     RAG is commonly considered a technique to alleviate the need for LLMs to read long contexts by chunking documents into pieces, which is how modern RAG systems operate. Therefore, technically speaking, long-context and RAG represent two different generation paradigms for LLMs. If your concern regarding xRAG involves multiple document chunks, please refer to our general response.
>
>
> Thank you again for your review. We hope that our response addresses your concerns and that you might consider raising the score of our paper.
>
> Best regards,
>
> Authors
>
> References:
>
> [1]  Lin, Xi Victoria, Xilun Chen, Mingda Chen, Weijia Shi, Maria Lomeli, Rich James, Pedro Rodriguez, Jacob Kahn, Gergely Szilvasy, Mike Lewis, Luke S. Zettlemoyer and Scott Yih. “RA-DIT: Retrieval-Augmented Dual Instruction Tuning.” *ArXiv* abs/2310.01352 (2023): n. pag.
>
> [2] Yu, Yue, Wei Ping, Zihan Liu, Boxin Wang, Jiaxuan You, Chao Zhang, Mohammad Shoeybi and Bryan Catanzaro. “RankRAG: Unifying Context Ranking with Retrieval-Augmented Generation in LLMs.” (2024).
>
> [3] Shao, Rulin, Jacqueline He, Akari Asai, Weijia Shi, Tim Dettmers, Sewon Min, Luke S. Zettlemoyer and Pang Wei Koh. “Scaling Retrieval-Based Language Models with a Trillion-Token Datastore.” (2024).
>
> [4] Lin, Sheng-Chieh, Akari Asai, Minghan Li, Barlas Oğuz, Jimmy J. Lin, Yashar Mehdad, Wen-tau Yih and Xilun Chen. “How to Train Your DRAGON: Diverse Augmentation Towards Generalizable Dense Retrieval.” *ArXiv* abs/2302.07452 (2023): n. pag.
>
> [5] Khattab, O. and Matei A. Zaharia. “ColBERT: Efficient and Effective Passage Search via Contextualized Late Interaction over BERT.” *Proceedings of the 43rd International ACM SIGIR Conference on Research and Development in Information Retrieval* (2020): n. pag.
>
> [6] Thakur, Nandan, Nils Reimers, Andreas Ruckl'e, Abhishek Srivastava and Iryna Gurevych. “BEIR: A Heterogenous Benchmark for Zero-shot Evaluation of Information Retrieval Models.” *ArXiv* abs/2104.08663 (2021): n. pag.
>
> [7] Muennighoff, Niklas, Nouamane Tazi, Loic Magne and Nils Reimers. “MTEB: Massive Text Embedding Benchmark.” *Conference of the European Chapter of the Association for Computational Linguistics* (2022).
>
> [8] Jin, Qiao, Bhuwan Dhingra, Zhengping Liu, William W. Cohen and Xinghua Lu. “PubMedQA: A Dataset for Biomedical Research Question Answering.” *Conference on Empirical Methods in Natural Language Processing* (2019).
>
> [9] Yang, Zi, Yue Zhou and Eric Nyberg. “Learning to Answer Biomedical Questions: OAQA at BioASQ 4B.” (2016).
>
> [10] Peng, Yifan, Shankai Yan and Zhiyong Lu. “Transfer Learning in Biomedical Natural Language Processing: An Evaluation of BERT and ELMo on Ten Benchmarking Datasets.” *BioNLP@ACL* (2019).
>
> [11] https://huggingface.co/datasets/ncbi/pubmed
>
> [12] https://blog.salesforceairesearch.com/sfr-embedded-mistral/

---

### Official Review · Reviewer_H8JS · 2024-07-14

**Soundness:** 3
**Presentation:** 4
**Contribution:** 3
**Rating:** 7
**Confidence:** 4

**Summary:**

This paper presents xRAG, a context compression method for Retrieval-Augmented Generation (RAG). Their key idea is to treat document embeddings from dense retrieval as features from a retrieval modality, which allows compressing retrieved documents into a single token. Experiments show that their method can achieve extreme compression while maintaining performance comparable with traditional RAG.

**Strengths:**

- The proposed xRAG method treats document embedding as a modality, and shows strong potential for efficient RAG systems. Specifically, the proposed method uses lightweight training strategy that consists of paraphrase pretraining and context-aware instruction tuning.
- xRAG demonstrates strong empirical results: it can achieve similar performance with traditional RAG methods on various QA datasets, and has the best performance among compression-based RAG methods. However, it still has limitations in tasks that require multi-step reasoning.
- The paper provides in-depth analysis of the method's behavior, including studies on robustness, effectiveness of different components, and failure cases.
- The writing is very clear and easy to understand.

**Weaknesses:**

- RAG is proposed to reduce the hallucination of language model generation, by letting them refer to the original context. It is still unclear why xRAG that compresses a document into a single token could still leave the factual knowledge intact, and the authors should discuss more on this.
- As the limitation paragraph mentions, the paper does not consider the case of retrieving multiple documents. Though the proposed method works for one-document case, it may not generalize to multiple-document case and may affect the training of the compression projector.

**Questions:**

Please refer to the previous section.

**Limitations:**

Yes, the paper addresses the limitations.

---

> ### Author Rebuttal · Authors · 2024-08-06
>
> Dear Reviewer,
>
> We greatly appreciate your time and effort in reviewing our paper. Here, we would like to address your concerns point by point:
>
> - **Question about why xRAG could compress a document chunk into a single token**
>
> Thank you for this insightful question. We believe this is the core of xRAG, and we want to explain it from two perspectives. First, it is surprising how much information a single embedding can contain. As demonstrated in [1], a single vector of shape 768 can recover 32 tokens with over a 90% exact match. In our experiments, we use an embedding of shape 4096 to cover a document chunk with an average of 180 tokens. Moreover, according to Claude Shannon's information theory, one classic estimate of the amount of information per English word is 11.82 bits per word [2]. This suggests that, in theory, we still have enough raw state to encode more tokens with a single embedding. Second, the overall performance is a joint effect from both Boost Rate and Resilience Rate (as defined in Section 6.1 of our paper) and we have to acknowledge that xRAG does not yet perform comparably with RAG in terms of Boost Rate. Improving the Boost Rate of xRAG while maintaining a high level of Resilience Rate—balancing external and internal knowledge—is a primary focus of our future work.
>
> - **About retrieving multiple documents**
>
> Please refer to our general response.
>
> Thank you again for your review. We hope that our response alleviates some of your concerns and that you might consider raising the score of our paper.
>
> Best regards,
>
> Authors
>
> [1] Morris, John X., Volodymyr Kuleshov, Vitaly Shmatikov and Alexander M. Rush. “Text Embeddings Reveal (Almost) As Much As Text.” *Conference on Empirical Methods in Natural Language Processing* (2023).
>
> [2] Claude Elwood Shannon. “Prediction and Entropy of Printed English.” (1951).

---

### Author Rebuttal · Authors · 2024-08-06

Dear Reviewers,

We sincerely appreciate your time and effort in reviewing our paper. We would like to address the concerns regarding the multiple document expansion of xRAG.

Our work represents a pioneering effort in efficient RAG with modality fusion, a research direction that has been acknowledged as both interesting and promising by the reviewers. In our current experimental setup, we have focused on the top-1 document in both training and evaluation to facilitate rapid iteration of data-centric experiments and for efficiency reasons. We view the expansion to multiple documents as an incremental improvement (1 to N) of the existing framework, which, while valuable, is less fundamental compared to our core innovation (0 to 1).

Adapting xRAG to a top-k documents setting presents several avenues for exploration, including:

1. The type of modality projector (e.g., MLP, Q-Former, or Perceiver).
2. Data selection and mixing strategies (e.g. data cluster with longer context)
3. Optimization for document relationship modeling and more.

To demonstrate xRAG's adaptability to multi-document scenarios, we have implemented a straightforward approach focusing primarily on the data perspective. Specifically, we upsampled summarization data, which typically includes longer documents, and divided these into chunks during our context-aware instruction tuning phase. We maintained other configurations as in the current xRAG implementation. The results of this approach are presented below.

While this naive implementation may not represent the optimal configuration for a multi-document setting in xRAG, as it primarily stems from the data aspect, it effectively showcases our framework's flexibility and extensibility to handle multiple documents.

|  | NQ | TriviaQA | WebQ | HotpotQA |
| --- | :---: | :---: | :---: | :---: |
| top1 | 39.1 | 65.7 | 39.4 | 34.0 |
| top3 | 41.4 | 67.3 | 41.1 | 35.3 |

Thank you once again for your valuable feedback.

Best regards,

Authors

---

### Author Response · Authors · 2024-08-12

Dear Reviewers,

We hope this message finds you well. As the end of the rebuttal phase is fast approaching (Aug 13 11:59pm AoE), we wanted to kindly request your feedback on the amendments and clarifications provided in the rebuttal. Your insights and critiques greatly contribute to enhancing the quality of our work.

We are more than willing to provide further clarifications or engage in additional discussions if needed.

Thank you once again for your time and consideration.

Sincerely,

Authors

---

### Decision · Program_Chairs · 2024-09-25

**Decision:**

Accept (poster)

**Comment:**

The paper presents a context compression approach for retrieval-augmented generation (RAG). The idea is to use document embeddings that were used for retrieval systems and train a light-weight adaptor that can integrate information from such embedding into LLMs. This compresses in-context document significantly, resulting in much more efficient model (with smaller FLOPS). Overall, the idea is pretty simple but well-executed and shows strong experimental gains.

The reviewers identified some limitations of the proposed approach - such as limited performance when handling multiple retrieved documents. While the approach shows efficiency gain in this setting, it comes with moderate performance drops. The new results from the discussion period should be added to the revised paper. With the advance of the long context model, extending this work to compressing long document setting will be necessary for future impact.